# Genomic Landscape and Prediction of Udder Traits in Saanen Dairy Goats

**DOI:** 10.3390/ani15020261

**Published:** 2025-01-17

**Authors:** Xiaoting Yao, Jiaxin Li, Jiaqi Fu, Xingquan Wang, Longgang Ma, Hojjat Asadollahpour Nanaei, Ali Mujtaba Shah, Zhuangbiao Zhang, Peipei Bian, Shishuo Zhou, Ao Wang, Xihong Wang, Yu Jiang

**Affiliations:** 1Key Laboratory of Animal Genetics, Breeding and Reproduction of Shaanxi Province, College of Animal Science and Technology, Northwest A&F University, Yangling 712100, China; yaoxiaoting@nwafu.edu.cn (X.Y.); lijiaxin280512@163.com (J.L.); fujiaqi92@163.com (J.F.); wxqwxq@nwafu.edu.cn (X.W.); 13369452684@163.com (L.M.); alimujtabashah@nwafu.edu.cn (A.M.S.); zhangzhuangbiao18@163.com (Z.Z.); bppisc@163.com (P.B.); zhoushishuo@nwafu.edu.cn (S.Z.); minerwang@nwafu.edu.cn (A.W.); 2Animal Science Research Department, Fars Agricultural and Natural Resources Research and Education Center, Agricultural Research, Education and Extension Organization (AREEO), Shiraz 7155863511, Iran; h.asadollahpour@agr.uk.ac.ir

**Keywords:** genotyping imputation, genomic prediction, heritability, breeding programs, machine learning

## Abstract

This study focused on advancing genetic selection and productivity in Saanen dairy goats, a key species in the dairy industry of Shaanxi, China, by exploring genomic prediction and the genetic architecture of udder traits. Using genome-wide association studies (GWAS) and phenotypic data from 635 Saanen dairy goats, heritability of udder traits was estimated to range from 0.13 to 0.32, and four candidate genes associated with these traits were identified. Three approaches—GBLUP, Kernel Ridge Regression (KRR), and Adaboost.RT—were compared to evaluate the accuracy and reliability of genomic prediction models for udder size traits. The machine learning models (KRR and Adaboost.RT) demonstrated superior performance, achieving up to 20% and 11% higher prediction accuracy than GBLUP, respectively, indicating their greater stability and reliability. These findings highlight the potential of combining GWAS and machine learning techniques to enhance genetic selection and productivity in dairy goats.

## 1. Introduction

Dairy goats, a key livestock species in Shaanxi, China, are valued for producing high-quality milk and dairy products. In 2023, milk production increased by 1.2%, with a 5% rise in average income for goat farms. In dairy goats, udder size is a key functional trait, directly influencing mechanical milking efficiency and longevity [1,2]. Given the considerable diversity in udder size across breeds, multiple dairy goat populations have utilized linear assessment techniques to incorporate these traits into breeding selection criteria [3,4]. Previously, dairy goat breeding selection focused mainly on production traits, which negatively impacted udder morphology, including increased udder depth and reduced teat verticality [5,6,7]. Thus, there is a demand to incorporate udder health traits into goat breeding programs, as it enhances adaptation to mechanical milking and reduces the risk of subclinical and clinical mastitis in the herd [8,9,10,11].

In recent years, some studies have estimated the genetic parameters of dairy goats on the basis of milk production traits [5,12]. However, the genomic evaluation and genome-wide association studies of udder size selection in the context of milk production improvement have received limited attention. This could be attributed to challenges in accurately measuring udder traits and variability in trait definitions across studies [13,14]. Through natural and artificial selection, distinct markers on domestic animal genomes have been established, and genomic selection has emerged as a crucial method for exploring breed specificity and selection characteristics in domestic ruminants [15]. Therefore, further work is necessary to assess genetic parameters, identify candidate genes, and estimate the accuracy of genomic prediction for udder size in goats, ultimately contributing to the improvement of goat milk production.

Machine learning (ML) involves the study of computer algorithms that enhance their performance automatically as they gain experience [16], which enables computer systems to efficiently execute tasks without explicit instructions, relying on pattern recognition and inference. Currently, ML has found extensive application across many areas in genetics and genomics, with ML proving particularly valuable for deciphering large genomic datasets [17]. It naturally follows that integrating ML with genotypic and phenotypic data allows for the calculation of genomic estimated breeding value (GEBV). Several studies have examined the application of ML in predicting genomic breeding values, comparing the effectiveness of ML techniques with commonly used methods such as Genomic Best Linear Unbiased Prediction (GBLUP), Bayes B, and Bayesian Lasso [18,19,20]. The outcomes of these techniques suggest that ML algorithms and similar methodologies primarily improve the accuracy of GEBV prediction [18,20,21], even necessitating a considerable amount of computing time in return for the disappointingly inadequate improvements. Evidently, there is an urgent need to explore novel approaches or strategies to apply ML algorithms in genomic prediction and enhance prediction accuracy.

The aim of the current research is to assess the genetic parameters governing udder size, identify genomic regions associated with udder traits through GWAS based on the resequencing data of dairy goats, and subsequently compare the reliability of different algorithms associated with these traits through genomic prediction. Our results are expected to provide useful insights into a deeper comprehension of the genetic basis of udder characteristics and will promote the future breeding of dairy goats to improve the udder traits through genomic prediction.

## 2. Materials and Methods

### 2.1. Phenotypes

Udder size traits for Saanen goats were collected at Shaanxi Aonike Dairy Co., Ltd. in Weinan, Shaanxi, China. This facility is notable as China’s first organic dairy farm for goat milk production, independently invested, constructed, operated, and managed. All clinically healthy dairy goats were categorized into three distinct age groups: 6 months, 1 year, and 2 years. As illustrated in Figure 1A, ‘a’ represents udder width, ‘b’ represents udder depth, and ‘c’ represents teat spacing. Images of these phenotypes were captured and subsequently analyzed using the computer software ImageJ (https://imagej.net/ij/), which enabled precise measurement and quantification of various phenotypic traits [22,23]. Import the image into ImageJ, set the scale based on the ruler visible in the image, and calculate the phenotypic length according to the defined scale (Appendix A). Ensure that the ruler and the phenotype being measured are located on the same plane. Phenotypic quality control was performed using R Software version 4.3.1, where incomplete or redundant records were removed by identifying duplicates using the ‘duplicated’ function. The dataset primarily comprised dairy-producing goats, accounting for approximately 82% of the records, while the remaining 18% were from non-dairy-producing goats. After data filtering, 86% of the available records were retained, resulting in a final dataset of 635 Saanen goats (Table 1).

### 2.2. Genotypes

Here, 635 Saanen dairy goats were genotyped using low-coverage whole-genome sequencing (lcWGS) on the MGI-2000/MGI-T7 sequencing platform in the PE150 mode. Genomic DNA was extracted from blood samples of Saanen dairy goats following the instructions of the TIANamp Blood DNA Kit (Tiangen Biotech Co. Ltd., Beijing, China). After confirmation of both phenotypes and genotypes, a total of 635 Saanen dairy goats were genotyped through sailegene at BaseNumber NGS [24], utilizing a high-performance GPU-accelerated variant calling tool for genome data analysis. Quality control was performed using bcftools-1.17, excluding SNP markers with a call rate < 0.90 to minimize missing genotype data, a minor allele frequency (MAF) < 0.05 to remove rare variants, monomorphic SNPs to eliminate non-informative markers, and SNPs deviating significantly (>0.15) from the Hardy–Weinberg equilibrium to ensure data integrity, resulting in the identification of 70,136 SNPs [25].

GLIMPSE2 utilizes a Gibbs sampler algorithm that alternates between two steps: imputing missing genetic data and phasing genetic variants [26]. It applies a modified version of the hidden Markov model (HMM) developed by Li and Stephens to perform imputation on lcWGS data. GLIMPSE2 was used to correct the called genotypes and achieve high-quality imputation of missing genotypes for the low-coverage sequencing data, which was appropriate for lcWGS in this study. Imputed genotypes for 635 animals were accessed, and GLIMPSE2 required 60 h of runtime while utilizing a maximum of 3 processes. Furthermore, individuals were excluded if the sample call rate was below 0.90 or if there were parent-progeny Mendelian conflicts. A final dataset consisting of 635 animals and 14,717,075 imputed SNPs was generated.

### 2.3. Estimation of Genetic Parameters

Genetic parameters and GEBV were estimated using multi-trait animal models with the AIREMLf90 and BLUPf90 program suites, respectively. On the basis of all available genomic data, genetic parameters were evaluated using a GBLUP approach with default setting and blending parameters for the genomic (G) relationship matrix. GBLUP has been used for to predict merit in livestock breeding, which can lead to more reliable predictions in genetic evaluations. The model for each trait contained the fixed effect of milk production status and birth year and the random effect of an animal additive genetic effect. Therefore, the multi-trait GBLUP model used for this examination is defined as follows:y1y2=I100I2μ1μ2+Z100Z2g1g2+e1e2
where y1y2 denotes the vector of observed values for traits I and II, representing corrected phenotypic values from different populations; The identity matrices $I_{1}$ and $I_{2}$ accompany the intercept vector μ1μ2 for the respective traits; The vector g1g2 contains the additive genetic effects, which follow a normal distribution N(0,G⊗M), where $M=\left[\begin{array}{cc} \delta_{g1}^{2} & \delta_{g12}^{2}\\ \delta_{g12}^{2} & \delta_{g2}^{2} \end{array}\right]$ represents the variance and covariance matrix of the genomic breeding values; The incidence matrices $Z_{1}$ and $Z_{2}$ link $g_{1}$ and $g_{2}$ to $y_{1}$ and $y_{2}$; The error vector e1e2 follows N(0,I⊗R), where $R=\left[\begin{array}{cc} \delta_{e1}^{2} & \delta_{e12}^{2}\\ \delta_{e12}^{2} & \delta_{e2}^{2} \end{array}\right]$ represents the residual variance and covariance matrix. The genetic correlation between traits is calculated as $\frac{\delta_{g12}^{2}}{\sqrt{\delta_{g1}^{2}\delta_{g2}^{2}}}$.

Furthermore, heritability for each given female udder trait was calculated using the following formula:$$h^{2}=\frac{V_{A}}{V_{A}+V_{E}}$$
where h^2^ represents heritability, V_A_ represents addictive genetic variance, and V_E_ represents environmental variance.

### 2.4. Kernel Ridge Regression

KRR is a non-linear regression strategy derived from ordinary least squares regression and ridge regression. It is often considered effective in uncovering the non-linear structure of data manifolds [27,28]. The difference between KRR and ridge regression lies in the utilization of the kernel trick by kernel ridge regression, enabling the definition of a higher-dimensional feature space, followed by the construction of the ridge regression model within this feature space. In this paper, the KRR prediction function can be expressed as follows:$$f\left(x\right)=k\prime {\left(K+\lambda I\right)}^{-1}y,$$
where K represents the Gram matrix with elements denoted as $K_{ij}=\Phi \left(x_{i}\right)\Phi \left(x_{j}\right),$ and k is a vector with components $k\left(x,x_{i}\right),$ with i represents the number of training individuals; I represents the identity matrix, and λ represents the ridge parameter. In this research, the kernel utilized to transform input data was chosen through the grid search method. The predictions of KRR were conducted using the scikit-learn package of Python, version 3.10.9. The regularization parameter α was fixed at 1.0 to balance bias and variance in the model.

### 2.5. Adaboost.RT

Adaboost.RT derives from Adaboost, which combines multiple base learners to establish a “committee” with the potential for superior predictive capability compared to any individual base learner [29]. Compared to other ensemble methods, the Adaboost.RT algorithm possesses the following outstanding characteristics, which are the reasons contributing to its status as a leading ensemble algorithm. In the Adaboost.RT algorithm, the accuracy of GEBV prediction for each individual relies on the thresholds Φ. If the relative error between the predicted and true values exceeds Φ, we classify this individual as mis-predicted. In this research, the Adaboost.RT regression model is formulated as follows:$$y=\sum_{t=1}^{M}(log\frac{1}{\varepsilon_{t}})f_{t}(x)/\sum_{t=1}^{M}(log\frac{1}{\varepsilon_{t}}),$$
where y represents the final predicted GEBV; $f_{t}(x)$ represents the predicted value of tth week learner; $\varepsilon_{t}$ represents the error rate of $f_{t}(x)$,$$\varepsilon_{t}=\sum_{i:{ARE}_{t}\left(i\right)>\varphi }D_{t}\left(i\right),$$
where ${ARE}_{t}\left(i\right)$ represents the absolute relative error of weak learner $f_{t}(x)$ for the ith sample; $D_{t}$ represents the distribution of $f_{t}(x)$’s weights; after training the weak learner $f_{t}(x)$, the weight distribution $D_{t}$ will be updated to $D_{t+1}$ for the next weak learner,$$D_{t+1}\left(i\right)=\frac{D_{t}(i)}{Z_{t}}\times \left\{\begin{array}{c} \varepsilon_{t}(if{ARE}_{t}\left(i\right)\leq \varphi ,\\ 1(if{ARE}_{t}\left(i\right)>\varphi \end{array}\right.$$
where $Z_{t}$ is a normalization element selected to ensure that $D_{t+1}$ forms a valid distribution.

To ensure acceptable computation time, Adaboost.RT utilized genome-derived relationships among samples as input data instead of direct genotypes [30]. The genome-derived relationship matrix G is computed using the following formula [31]:$$G=\frac{MM^{\prime }}{\sum_{l=1}^{m}2p_{j}q_{j}},$$
where M represents a n × m matrix (n indicates the count of individuals, m indicates the number of markers), the factors of column j in M are $0-2p_{j}$, $1-2p_{j}$, and $2-2p_{j}$ for genotypes $A_{1}A_{1}$, $A_{1}A_{2}$, and $A_{2}A_{2}$, respectively; $q_{j}$ represents the allele frequency of $A_{1}$ at locus j, and $p_{j}$ represents the allele frequency of $A_{2}$ at locus j. In this study, Adaboost.RT was applied in Python V3.10.9 with the scikit-learn package. The parameter of n_estimators was set to 100, specifying the total number of base learners employed in the boosting process.

### 2.6. Validation of Prediction Efficiency

To evaluate the GBLUP prediction efficiency, we adapted a 10-fold cross-validation test for each trait in the dairy goat population, which could benefit large training groups and mitigate high variation in genomic prediction accuracies [32,33]. For this purpose, ten training and ten validation groups were selected from the dairy goat population. The records were randomized, and ten equal subsets were made. The division into training and validation groups followed a pattern where each training group comprised nine out of ten subsets, leaving the tenth subset designated as the validation group. As a result, ten training and validation sets were established for each trait, which were employed to estimate genetic parameters and assess the accuracy of genomic predictions. The efficiency of genomic prediction was measured as the correlation coefficient between breeding values and phenotypic values adjusted for fixed effects. On the basis of 10-fold cross-validation, the accuracy of genomic prediction for each trait was determined by averaging the accuracies across the ten validation groups [34].

### 2.7. GWAS

An extensive GWAS was conducted on the basis of three sets of phenotypic data and genotypic data from 14,717,075 common SNPs with minor allele frequency (MAF) > 0.05. The missing information in the goat SNP genotype dataset was imputed with GLIMPSE2 [35,36]. The association analysis was conducted with a Genome-wide Efficient Mixed Model Association program (GEMMA). Milk production status and birth year were used as fixed effects, while principal components (PCA) were used as covariates. The suggestive significance threshold was approximated to be around *p* = 10^−6^, which allows for a more balanced approach to identifying potential genetic associations [37,38]. Genome-wide linkage disequilibrium (LD) attenuation analysis determined 50 kb as the optimal linkage distance, as it provides an appropriate balance between maintaining sufficient genetic correlation and minimizing the influence of distant loci [2,39]. This distance was subsequently used as the window size in further analyses.

## 3. Results

### 3.1. Imputation of Missing Genotypes

Previous studies have demonstrated the effectiveness of QUILT with lcWGS data; however, its performance slows down with larger reference panel sizes [40]. GLIMPSE2 imputation software was assessed for both speed and accuracy of imputation to enhance imputation efficiency [41]. Considering the absence of a large publicly available reference panel of goats, we have established a new reference panel consisting of 1232 goats. The sequencing depth of goats lcWGS was around 1.95×, and there are 14,717,075 high-quality SNPs in the final genomic dataset after imputation. Moreover, imputation accuracy was also investigated using a recently referenced panel for the lcWGS in Saanen dairy goats. Twenty randomly selected samples were subjected to high-coverage resequencing (10×) to generate the true set, which was then compared to corresponding imputed samples. We estimated the imputation accuracy through Pearson correlation coefficient (R^2^) and concordance. Our findings demonstrated an average accuracy of 97.3% and a correlation rate of 92.8% between the imputed dataset and the ground truth dataset, indicating that the established reference panel exhibits excellent imputation accuracy for lcWGS data in Saanen dairy goats, thus making it suitable for future studies (Figure 2).

### 3.2. Genetic Parameters

The heritability estimates for the udder traits ranged from low to moderate, spanning between 0.13 to 0.32 for teat spacing and udder depth in Saanen (Table 2). Specifically, the estimate consistently indicated a higher heritability for udder depth compared to both udder width and teat spacing. The variance components for these traits revealed that the genetic variance contributed significantly to the total variation, with udder depth (35.07) showing a higher proportion of genetic variance compared to udder width (4.23) and teat spacing (1.40). The variance of the residual effect, representing unexplained variability and random error, was relatively higher for udder depth (74.52) compared to udder width (22.21) and teat spacing (9.37).

The correlation estimates for udder size traits are substantiated by a phenotypic correlation among them exceeding 0.5. Specifically, the correlation between udder width and udder depth is 0.72, the correlation between udder width and teat spacing is 0.80, and the correlation between udder depth and teat spacing is 0.51 (Table 2). Regarding the genetic correlation among udder traits, we observed a range from 0.45 to 0.79, while the high standard error may be due to the limitation in sample size.

The heritability values are shown on the diagonal, measuring the fraction of phenotype variability that can be attributed to genetic variation for each trait. The genetic correlations, which are displayed above the diagonal, reflect the relationships between the genes controlling the two traits. Phenotypic correlations, found below the diagonal, reflect the overall correlation between traits, combining both genetic and environmental influences. Standard errors are given in brackets.

### 3.3. Genomic Prediction

Compared to ML, the efficiency of genomic prediction is depicted in Figure 3 for udder traits, considering the reference population size. The accuracy of GEBV for udder width was 0.18 when considering a reference population of about 635 individuals. The accuracy of prediction udder depth was slightly higher (0.29), while the accuracy of GEBV for teat spacing was lower (0.15).

An advantage was found in applying ML methods (KRR and Adaboost.RT) over GBLUP for udder width, udder depth, and teat spacing in Saanen dairy goats (Figure 3). For udder width, the prediction accuracy of KRR, Adaboost.RT, and GBLUP were 0.22 ± 0.07 (95% CI: [0.18, 0.26]), 0.20 ± 0.06 (95% CI: [0.16, 0.23]), 0.18 ± 0.10 (95% CI: [0.12, 0.24]). The improvement achieved by ML methods over GBLUP was 22% and 11%. For udder depth, the prediction accuracy of KRR, Adaboost.RT, and GBLUP were 0.24 ± 0.05 (95% CI: [0.21, 0.27]), 0.23 ± 0.07 (95% CI: [0.19, 0.27]), and 0.22 ± 0.08 (95% CI: [0.17, 0.27]), and the improvement achieved by ML methods over GBLUP was 9% and 5%. Similarly, compared to GBLUP (0.17 ± 0.06) (95% CI: [0.13, 0.21]), the accuracy of KRR (0.22 ± 0.07) (95% CI: [0.18, 0.26]) and Adaboost.RT (0.20 ± 0.06) (95% CI: [0.16, 0.24]) has 29% and 18% improvement for teat spacing. The accuracy of Adaboost.RT is similar to that of KRR, both demonstrating the highest level of predictive accuracy.

### 3.4. GWAS

We phenotyped a GWAS panel comprising 635 diverse accessions raised in Saanen dairy goats farm in China (Figure 1). We sequenced the GWAS panel with approximately 1.95-fold coverage, generating a total of 3.39 terabases of raw sequence data. Using the genotypic data of 14,717,075 high-quality SNPs with MAF > 0.05, we identified 101 associated loci—12 for udder width-related traits, 19 for udder depth, and 70 for teat spacing (Figure 4 and Appendix A)—with a suggestive threshold (*p* < 1 × 10^−6^ in a mixed model). Among these 101 associated loci, three are co-occurring between udder width and udder depth, while one locus is shared between udder width and teat spacing (Appendix A). There were 32 unique annotated positional candidate genes identified in all analyses, comprising 78% protein-coding genes and 22% nonprotein-coding genes, specifically long noncoding RNA (Appendix A).

Additionally, gene annotation was conducted to enhance understanding of the biological mechanisms and genetic architecture of udder size traits in Saanen dairy goats. The potential candidate genes for udder traits were primarily associated with tissue and organ morphology. Among them, the *GJB5* gene on chromosome 3 was associated with udder width and udder depth in Saanen dairy goats. The most significant effects for udder width were identified within the intergenic region of the *GJB5* gene at position 9,872,130 bp, exhibiting the same level of significance as observed for udder depth (Figure 4, Appendix A). Meanwhile, *PKN2* is shared between udder width and teat spacing. Three potential genes (*GJB5*, *PKN2*, and *KTN1*) were identified for udder width on chromosomes 3 and 10, explaining 6%, 4%, and 4% of the phenotypic variation, respectively. However, only one major candidate gene (*GJB5*) was identified for udder depth, which explained 6% of the phenotypic variation. Additionally, two potential genes were identified for teat spacing (*PKN2* and *NAV3*), located on chromosomes 3 and 5, explaining 6% and 4% of the phenotypic variation, respectively. Detailed information on candidate genes based on traits is provided in Appendix A.

## 4. Discussion

Due to the rapid reduction of sequencing cost and the advancement of methods and algorithms for managing whole-genome sequencing data, there is an increasing focus on whole-genomic variations rather than the restricted set of variants detected by SNP arrays [42,43]. Low-coverage sequencing coupled with the imputation of missing genotypes has been suggested to be a cost-effective method for acquiring genotypes of whole-genome variants [44,45,46]. A sequencing depth of 1.95× is more cost-effective than higher depths, making it suitable for large-scale studies with budget constraints. While lower depth may reduce imputation accuracy, increasing the sample size can compensate by providing more genetic variation, thus enhancing the statistical power of downstream analyses. The imputation performance (e.g., accuracy, SNP discovery, and computational time) plays a crucial role in the effectiveness of the lcWGS’s data, which is influenced by various factors, including the imputation method, sequencing depth, sample size, and the number of sequenced individuals, as well as the availability and size of the reference panel. According to previous studies, GLIMPSE2 has been considered for genotype imputation due to its strong performance (accuracy > 0.90), even for SNPs with a MAF below 0.01 [40,41]. Consistent with previous research findings, our results indicated that, with a reference panel of considerable size (1232), the imputation accuracies remained consistently high (exceeding 0.95), thus facilitating subsequent analyses.

In the present study, the estimated heritability values for udder size traits ranged from 0.13 to 0.32, consistent with findings reported in recent studies on livestock [47,48]. Previous studies on dairy goats have reported a heritability estimate of 0.25 for udder-type traits, suggesting moderate genetic influence and supporting the potential for selection-based improvements in udder morphology within this species [11]. The variance components analysis showed that udder depth had a higher genetic variance (35.07) compared to udder width (4.23) and teat spacing (1.40), suggesting a stronger genetic determination for udder depth. However, udder depth also had a higher residual variance (74.52), indicating that environmental factors or errors contribute significantly. In contrast, teat spacing had the lowest residual variance (9.37), implying that its variability is more genetic in nature. These results highlight the need to consider both genetic and environmental factors when evaluating udder traits. Considering the limited investigation into goat udder size, we referred to studies on genomic prediction for cattle and observed a comparable heritability estimate to current research [49]. Similar to our research, Schmidtmann et al. applied linear models and reported considerably moderate heritability for udder depth in German Holstein cattle [49]. Moderate heritability estimates for udder traits indicate a complex genetic basis. In breeding programs, a multi-generational accumulation strategy can be employed to achieve gradual improvements through incremental selection. Furthermore, all genetic correlations among udder size traits were positive and high, indicating a strong genetic relationship between these traits, which are likely influenced by common genetic factors. The high phenotypic correlation (>0.5) among traits suggests shared genetic influences, which has important implications for selection strategies. Nonetheless, given the heritability of these traits and the breed’s variability, producers have the opportunity to select for larger and tighter suspensions, thus improving udder quality within their herds.

The original design of this study is to utilize an ensemble learning algorithm to establish a genomic prediction model, enhancing its accuracy in predicting GEBVs for critical economic traits of Saanen dairy goats. The outcomes of comparing the predictive performance of GBLUP with KRR and Adaboost.RT indicate that both ML methods enhance the prediction accuracy of GEBVs, with KRR showing particularly notable improvements. In contrast to previous research findings, most relied on a single ML algorithm for GEBV estimation without significant improvement [18,50]. However, Liang et al. suggested that the robust reliability observed in the predicted GEBVs obtained through the ensemble-learning method highlighted the significance of ML algorithms in improving the prediction accuracy of GEBVs. They compared the reliability of KRR, Adaboost.RT, and GBLUP methods, demonstrating that machine learning methods outperformed GBLUP, with average improvements of 14.9% and 14.4% for KRR and Adaboost.RT, respectively [51]. Consistent with previous research, our findings suggest that ML methods demonstrate excellent reliability in predicting GEBVs, achieving average improvements of over 10% and further confirming the superiority of nonlinear models in genomic prediction. While GBLUP assumes linear genetic effects, KRR and AdaBoost.RT can capture complex interactions and non-linearities in the genetic architecture. This flexibility allows ML methods to provide more accurate predictions, especially when dealing with large, high-dimensional genomic data. This study focuses on predictive accuracy, but it is important to acknowledge the potential computational costs associated with ML methods compared to traditional models like GBLUP. Future research could investigate strategies to optimize computational efficiency to facilitate broader practical applications.

By conducting GWAS on imputed whole genome sequences in dairy goats and selecting candidate variants, this study successfully identified 32 unique annotated positional candidate genes and found that *GJB5*, *PKN2*, *KTN1,* and *NAV3* were related to udder size traits. *GJB5* is classified among genes linked to pluripotency, suggesting its potential role in modulating the functions and traits of diverse cell types [52]. Within mammary tissue, *GJB5* might impact mammary gland cells’ proliferation, differentiation, and interconnection, thereby influencing udder size and shape [53]. *PKN2* has been shown to interact with various proteins, including *APPL1* and *AKT*, forming complexes that regulate myocyte differentiation by modulating *AKT* activity and *MyoD*-mediated signaling pathways [54]. Consequently, it is plausible that *PKN2* plays a role in regulating udder size traits through its involvement in cellular differentiation and muscle development processes, potentially influencing mammary tissue growth and development. *KTN1* is a membrane-associated protein that interacts with endoplasmic reticulum (ER) proteins. Previous studies have shown its preference for binding to polyglutamylated microtubules surrounding the nucleus and specialized cytoskeletal components [55]. This suggested that *KTN1* may influence mammary tissue growth and development by modulating the localization and morphology of the ER within cells. Besides, research has indicated that *NAV3* localizes to the plus ends of microtubules and enhances their polarized growth upon induction by growth factors [56,57]. Therefore, NAV3 likely plays a role in regulating cell migration and microtubule dynamics, promoting sustained directional movement while limiting the mobility of specific cells. The identification of genes associated with udder traits offers valuable genetic markers for incorporation into breeding programs. These markers can be utilized in genomic selection to identify animals with superior udder traits early, thereby enhancing selection efficiency and accelerating genetic progress. Nevertheless, functional validation of the identified candidate genes using molecular biology techniques is essential for elucidating their specific roles in mammary gland development. Gene-editing technologies, such as CRISPR-Cas9, can be employed to knock out or overexpress candidate genes in mammary cells or animal models, enabling the assessment of phenotypic changes in gland development. Once the functional roles of these genes are validated, their genetic variants (e.g., SNPs) can be integrated into a genomic selection index. Associating these markers with specific udder traits will enable more accurate prediction of an animal’s genetic potential, facilitating more efficient selection for superior udder traits and improving breeding outcomes.

Building on these considerations, it is clear that while our study has made a significant contribution to understanding genomic variations associated with udder size in Saanen dairy goats, there are several areas that warrant further attention and refinement. Despite the robust imputation accuracy achieved in this study with a reference panel of 1232 individuals, it is important to acknowledge the limitations imposed by the sample size and the representativeness of the reference panel. Our use of linear models for heritability estimation may not have fully captured the subtle effects of environmental factors on udder size traits, such as nutritional regimens and management practices, which could be a valuable avenue for future investigation. Future research should aim to expand the sample size and diversity, incorporating more goat breeds and varying environmental conditions to enhance the generalizability of findings and the precision of heritability estimates. Finally, the continued exploration and development of ML algorithms will be instrumental in enhancing genomic prediction models for breeding Saanen dairy goats.

## 5. Conclusions

This study found moderate heritability for udder size traits in Saanen dairy goats, indicating a significant genetic influence on these traits. The GWAS identified associated loci, suggesting strong selection signatures for udder size traits. The model evaluation showed that machine learning algorithms, such as KRR and Adaboost.RT, outperformed linear methods like GBLUP in predictive accuracy, highlighting the potential of machine learning models to enhance the efficiency of genomic prediction in dairy goat breeding. Overall, this research provides valuable insights for improving udder-related economic traits through genomic prediction strategies in dairy goats.

## Figures and Tables

**Figure 1 animals-15-00261-f001:**
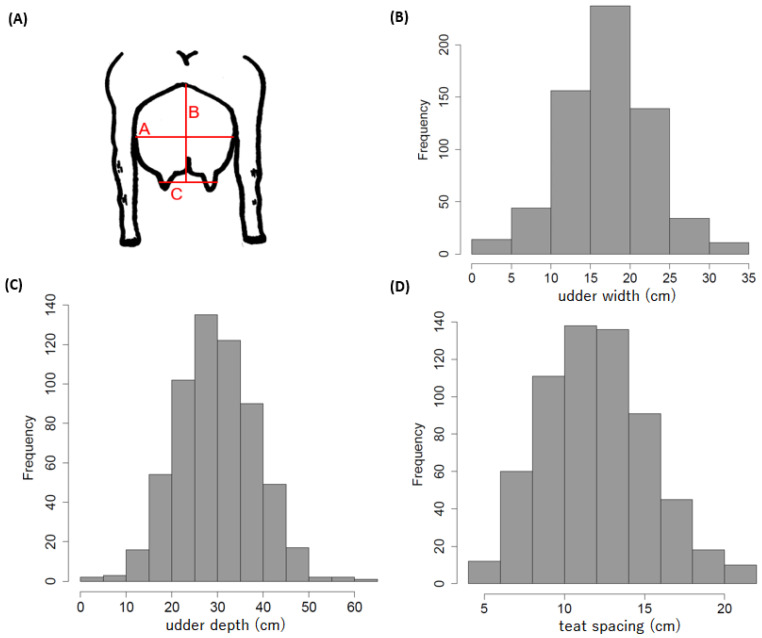
Frequency distribution of phenotypic variation of three udder traits in this study. (**A**) illustrates the phenotypic representation of the measured udder traits, where A corresponds to udder width, B corresponds to udder depth, and C corresponds to teat spacing. (**B**–**D**) represent the frequency distributions for udder width, udder depth, and teat spacing, respectively.

**Figure 2 animals-15-00261-f002:**
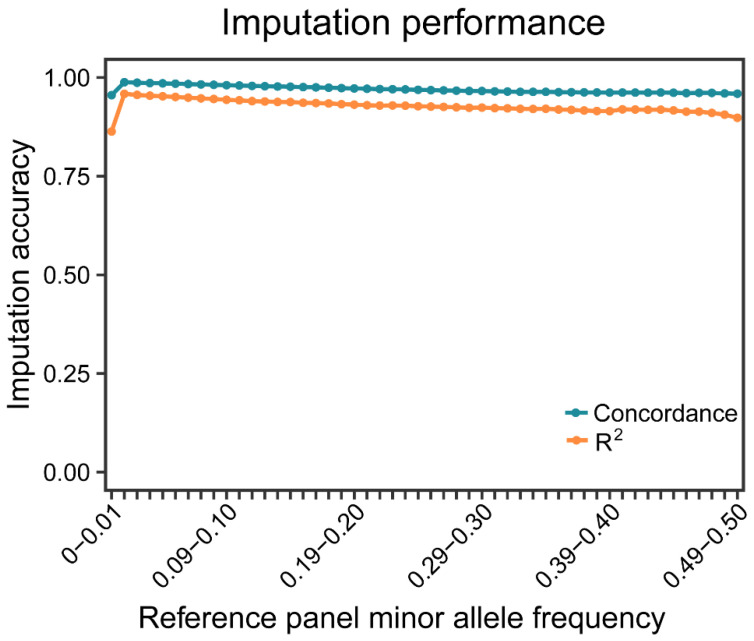
The performance for imputing lcWGS data. The histogram of imputed variants, concordance, and Pearson correlation coefficients for our reference panel.

**Figure 3 animals-15-00261-f003:**
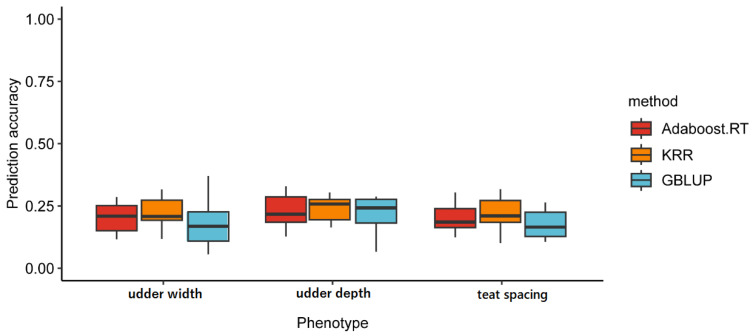
Prediction accuracy across three kinds of udder size traits using different genomic prediction methods.

**Figure 4 animals-15-00261-f004:**
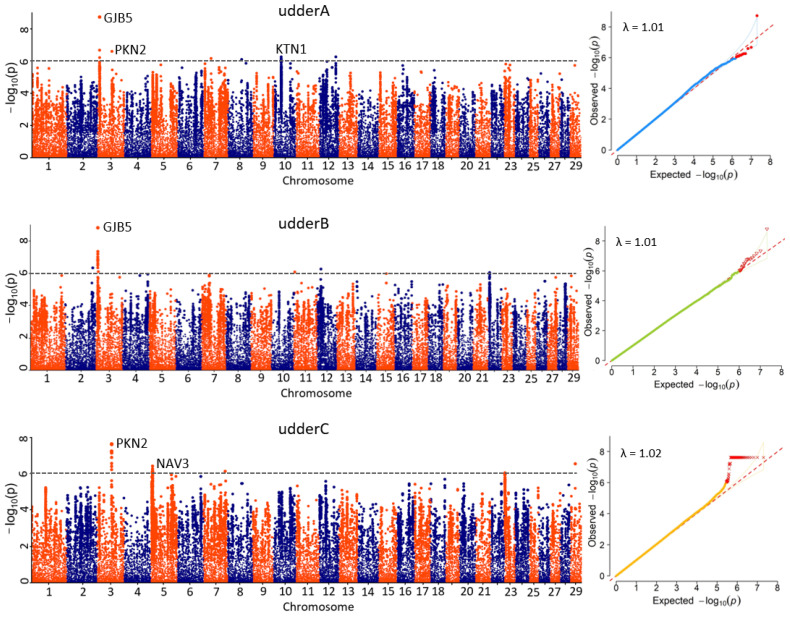
Manhattan plots and QQ plots of GWAS for udder size traits of Saanen dairy goats, using the lcWGS in 635 Saanen dairy goats. On the left side of the graph, *p*-values (−log10 (*p*)) for each SNP are plotted on the *Y*-axis, with chromosomes ranging from 1 to 29 displayed on the *X*-axis. The dotted line indicates the threshold of genome-wide significance. On the right side of the graph, the horizontal axis displays the −log10 transformed expected *p*-values, while the vertical axis represents the −log10 transformed observed *p*-values. These values are plotted against the expected *p*-values under the null hypothesis of no association, include red dots that represent genetically variant sites which are statistically significant.

**Table 1 animals-15-00261-t001:** Descriptive statistics for the udder size.

Trait	Udder Width	Udder Depth	Teat Spacing
Number of animals	635	635	635
Maximum	34.55 cm	61.24 cm	21.93 cm
Minimum	3.39 cm	3.75 cm	4.13 cm
Mean	17.15 cm	29.65 cm	11.96 cm
Standard deviation	5.48 cm	8.59 cm	3.33 cm
Coefficient of variation	0.32	0.29	0.28

**Table 2 animals-15-00261-t002:** Heritability (on the diagonal), genetic correlations (above the diagonal), and phenotypic correlations (below the diagonal) for udder size traits of dairy goats.

Trait	Udder Width	Udder Depth	Teat Spacing
udder width	0.16 (0.003)	0.79 (0.018)	0.70 (0.049)
udder depth	0.72 (0.001)	0.32 (0.004)	0.45 (0.014)
teat spacing	0.80 (0.001)	0.51 (0.001)	0.13 (0.003)

## Data Availability

The raw data has been uploaded to NCBI but is currently not publicly available; further inquiries can be directed to the corresponding authors.

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
