# Peer review of "Genomic Landscape and Prediction of Udder Traits in Saanen Dairy Goats"

_animals, 2025, doi:10.3390/ani15020261_

Round 1
Reviewer 1 Report
Comments and Suggestions for Authors
General Comments
The manuscript presents a well-structured and informative study on the genetic landscape and genomic prediction of udder traits in Saanen dairy goats. The integration of GWAS and machine learning methods for genomic prediction is innovative and highly relevant to the field. However, several areas require attention to enhance clarity, accuracy, and scientific rigor. The Materials and Methods section should be more detailed, particularly regarding the phenotypic data collection process, which might need to be rewritten or recalculated more carefully.Overall, the manuscript is very interesting and provides valuable insights, but the purpose of applying machine learning methods seems somewhat disconnected from the rest of the study, and the comparison between the machine learning approaches and traditional models is not adequately highlighted.
Title and Abstract
- Title (Line 1):
- The title is informative but could be more concise. Consider revising to "Genomic Landscape and Prediction of Udder Traits in Saanen Dairy Goats."
- Abstract (Lines 17–47):
- The abstract is well-written but overly technical. Simplify for a broader audience without losing key details.
- Typo: "machine learning models" (Line 21) should be corrected to "machine learning models."
- The term "up to 20% higher prediction accuracy" is vague. Include specific metrics or comparative results for clarity.
Introduction (Lines 53–97)
- Relevance (Lines 54–65): The introduction effectively highlights the importance of udder traits for dairy goats. However, some sentences are overly verbose (e.g., Line 65). Condense where possible.
- Literature Gap (Lines 68–70): Expand on why genomic evaluation of udder traits has been less explored compared to milk production traits. This provides stronger justification for the study.
Materials and Method
- Phenotypic Data (Lines 98–114):
- Clarify the software and specific steps used in ImageJ for phenotypic analysis (Line 104). This ensures reproducibility. Please also report some real images.
- More detail about animnals must be included: genetic background, age, etc.
- More traits in my opinion should have been included in particular some nipples measurement and angles
- Figure 1 and table 1: minimum of udder A and B seem very strange; it is possible the udder width is only 3.39 cm? It seems that there are some wrong values
- Line 107: how R was used, what statistical strategy? R is only a platform not a method.
- Genotypic Data (Lines 115–136):
- Explain why specific thresholds (e.g., MAF > 0.05, call rate < 0.90) were chosen for SNP filtering (Lines 131–133).
- Add details about WGS analysis, for example low-coverage stand for? PE? Platform?
- It is unclear whether "Sailegene" is the name of the genotyping technology, a commercial kit, or software used for data processing. A brief description or citation would help clarify this term for the reader.
- The sentence on "Gibbs sampler algorithm" (Line 122) is too technical. Simplify for a broader audience.
- Earlier in the manuscript (Line 31), it mentions 14,717,075 SNPs after imputation, but here, only 70,136 SNPs are mentioned. Clarify whether 70,136 SNPs represent raw data before imputation, or a subset used for analysis. This discrepancy could confuse readers.
- Strategy that leads to make two different genotyping on the same samples (?) is unclear. Better have been to make different set of samples and then use imputation to get more coverage on low density data- anyway this part should be better explained and justified.
- Genetic Parameters (Lines 137–157):
- Provide more detail on why certain models (e.g., GBLUP) were selected. Highlight their advantages over alternatives.
- In the model kidding date or kidding number could be considered?
- A covariate or fixed term category for milk production in kg should be considered or in some cases a bitrait model including the morphological trait and milk could help in estimation accuracy.
- The parity class or kidding date should be considered.
- What does Milk production status stand for?
- Pedigree has been included. If yes details must be provided
- Machine Learning Models (Lines 159–193):
- Clarify the specific hyperparameters used for KRR and Adaboost.RT (e.g., Line 171). This ensures reproducibility.
- Define "weak learners" in the context of Adaboost.RT (Line 175).
- Define better why use Kernel Ridge Regression and Adaboost.RT. Make a brief introduction in the introduction section and justify why you are going to do this comparison. The rationale beyond the use these 3 methodologies is not clearly explained.
- GWAS (Lines 216–226):
- Justify the choice of 50 kb as the optimal linkage distance (Line 225).
- As for Gibbs fixed term combination should be revised.
Results
- Genetic Parameters (Lines 247–262):
- The high phenotypic correlation (>0.5) among traits is significant. Discuss its implications for selection strategies.
- Genomic Prediction (Lines 267–285):
- The claim that ML methods improve GEBV accuracy by 22% and 11% (Line 278) is promising. Include confidence intervals to support these results.
- GWAS (Lines 286–318):
- The identification of candidate genes is a major strength. However, the discussion on functional roles (e.g., Lines 305–317) lacks depth. Add more biological context or references.
Discussion
- Heritability Estimates (Lines 334–344):
- Compare the heritability estimates with similar studies on goats or other livestock to contextualize findings and report in the text for better readability.
- The impact of environmental factors (Line 387) is mentioned briefly. Suggest conducting future studies to isolate these effects.
- Machine Learning (Lines 345–356):
- Discuss the computational cost of ML methods compared to traditional models like GBLUP. This is critical for practical applications.
- Candidate Genes (Lines 357–379):
- The functional roles of genes like GJB5 and PKN2 (Line 360) are intriguing but speculative. Suggest experimental validation for future studies.
Figures and Tables
- Figure 1 (Page 3):
- Add clear labels or annotations for each udder trait to improve comprehension.
- Figure 3 (Page 8):
- The box plots are informative but lack clarity on error margins. Include confidence intervals or standard deviations.
- Figure 4 (Page 9):
- The Manhattan plots are visually clear but lack a legend explaining the significance threshold.
- Table 2 (Page 7):
- The use of standard errors in parentheses is appropriate. However, explain this format in the legend for clarity.
Typos and Errors
- Line 323: "suggested as" → "suggested to be."
- Line 381: "significant contributions to" → "significant contribution toward."
Author Response
General Comments
The manuscript presents a well-structured and informative study on the genetic landscape and genomic prediction of udder traits in Saanen dairy goats. The integration of GWAS and machine learning methods for genomic prediction is innovative and highly relevant to the field. However, several areas require attention to enhance clarity, accuracy, and scientific rigor. The Materials and Methods section should be more detailed, particularly regarding the phenotypic data collection process, which might need to be rewritten or recalculated more carefully.Overall, the manuscript is very interesting and provides valuable insights, but the purpose of applying machine learning methods seems somewhat disconnected from the rest of the study, and the comparison between the machine learning approaches and traditional models is not adequately highlighted.
Title and Abstract
Title (Line 1):
- The title is informative but could be more concise. Consider revising to "Genomic Landscape and Prediction of Udder Traits in Saanen Dairy Goats."
Author Response: Thanks for your suggestion. We have revised the title to “Genomic Landscape and Prediction of Udder Traits in Saanen Dairy Goats”. (line 2-3)
Abstract (Lines 17–47):
- The abstract is well-written but overly technical. Simplify for a broader audience without losing key details.
Author Response: We thank the reviewer for pointing this out. The abstract has been simplified in the revised manuscript; kindly review it for accuracy. (line 28-42)
- Typo: "machine learning models" (Line 21) should be corrected to "machine learning models."
Author Response: We sincerely apologize for the error in our writing, which has been fixed in lines 23-24.
- The term "up to 20% higher prediction accuracy" is vague. Include specific metrics or comparative results for clarity.
Author Response: Thank you for your reminder. We have revised this sentence in the updated manuscript. (lines 23-24)
Introduction (Lines 53–97)
- Relevance (Lines 54–65): The introduction effectively highlights the importance of udder traits for dairy goats. However, some sentences are overly verbose (e.g., Line 65). Condense where possible.
Author Response: Thank you for your suggestion. We have revised this section to make it more concise. (lines 47-57)
- Literature Gap (Lines 68–70): Expand on why genomic evaluation of udder traits has been less explored compared to milk production traits. This provides stronger justification for the study.
Author Response: We thank the reviewer for pointing this out. We have expanded the explanation of why genomic evaluation of udder traits has been less explored and added relevant references in the updated manuscript. (lines 59-62)
Materials and Method
Phenotypic Data (Lines 98–114):
- Clarify the software and specific steps used in ImageJ for phenotypic analysis (Line 104). This ensures reproducibility. Please also report some real images.
Author Response: I appreciate your recommendation. We have added the specific steps of ImageJ and report the real images in the figure S1. (lines 100-103)
- More detail about animnals must be included: genetic background, age, etc.
Author Response: Thanks for your suggestion. We have added more details about the animals in lines 95-97.
- More traits in my opinion should have been included in particular some nipples measurement and angles
Author Response: We appreciate the reviewer’s suggestion regarding additional teat measurements and angles. Due to current resource constraints, we were unable to include these measurements in the present study. However, we plan to incorporate these traits in future research as our resources allow.
- Figure 1 and table 1: minimum of udder A and B seem very strange; it is possible the udder width is only 3.39 cm? It seems that there are some wrong values
Author Response: Thank you for your reminder. We confirm that the value of 3.39 cm is accurate. The goats in our study include individuals that have not yet completed their reproductive cycle, and as such, their udders have not fully developed.
- Line 107: how R was used, what statistical strategy? R is only a platform not a method.
Author Response: Thank you for your valuable comment. In the revised manuscript, we have added more details in lines 103-105.
- Genotypic Data (Lines 115–136):
- Explain why specific thresholds (e.g., MAF > 0.05, call rate < 0.90) were chosen for SNP filtering (Lines 131–133).
Author Response: Thank you for your comment. The thresholds for SNP filtering were chosen based on established practices in genetic data analysis to ensure data quality and statistical reliability. SNPs with a call rate < 0.90 were excluded to minimize the impact of missing genotype data and ensure reliable genotyping across samples. SNPs with a minor allele frequency (MAF) < 0.05 were removed to avoid rare variants, which may have limited statistical power and could introduce noise into the analysis. We have added this explanation in the revised manuscript. (lines 122-127)
- Add details about WGS analysis, for example low-coverage stand for? PE? Platform?
Author Response: Thank you for your suggestion. We have added this information in line 117.
- It is unclear whether "Sailegene" is the name of the genotyping technology, a commercial kit, or software used for data processing. A brief description or citation would help clarify this term for the reader.
Author Response: Thank you for your suggestion. We have added this information in lines 121-122.
- The sentence on "Gibbs sampler algorithm" (Line 122) is too technical. Simplify for a broader audience.
Author Response: Thank you for your suggestion. We have revised the sentence regarding the 'Gibbs sampler algorithm' to make it more accessible to a broader audience. (lines 128-131)
- Earlier in the manuscript (Line 31), it mentions 14,717,075 SNPsafter imputation, but here, only 70,136 SNPs are mentioned. Clarify whether 70,136 SNPs represent raw data before imputation, or a subset used for analysis. This discrepancy could confuse readers.
Author Response: Thank you for your suggestion. To clarify, the 14,717,075 SNPs mentioned earlier in the manuscript refer to the imputed SNPs. The 70,136 SNPs mentioned here represent a subset of SNPs used for analysis after quality control and filtering. We have updated the manuscript to provide a clearer distinction between these two sets of SNPs. (lines 127, 136-137)
- Strategy that leads to make two different genotyping on the same samples (?) is unclear. Better have been to make different set of samples and then use imputation to get more coverage on low density data- anyway this part should be better explained and justified.
Author Response: Thank you for your suggestion. We have provided detailed explanations and have also assessed the accuracy of imputation. This approach ensures that our dataset has greater reliability and completeness. (lines 127, 136-137)
- Genetic Parameters (Lines 137–157):
- Provide more detail on why certain models (e.g., GBLUP) were selected. Highlight their advantages over alternatives.
Author Response: Thank you for your reminder. We have added more detail in the revised manuscript. (lines 139-141)
- In the model kidding date or kidding number could be considered?
Author Response: Yes, kidding date and kidding number can be considered in the GBLUP model, but they are not necessary. And we have not collected accurate data regarding these factors, so they are not included in our analysis.
- A covariate or fixed term category for milk production in kg should be considered or in some cases a bitrait model including the morphological trait and milk could help in estimation accuracy.
Author Response: Yes, incorporating a covariate or fixed term category for milk production in kilograms would be preferable for enhancing the accuracy of our model. However, the goat farms are unable to provide us with accurate milk production for each individual.
- The parity class or kidding date should be considered.
Author Response: I apologize, but we are unable to obtain this information.
- What does Milk production status stand for?
Author Response: Milk production status indicates whether the goat is currently producing milk.
- Pedigree has been included. If yes details must be provided
Author Response: In this study, we didn’t use pedigree information, all analyses were based on genomic data.
- Machine Learning Models (Lines 159–193):
- Clarify the specific hyperparameters used for KRR and Adaboost.RT (e.g., Line 171). This ensures reproducibility.
Author Response: Thank you for your reminder. We have clarified the specific hyperparameters used for both KRR and Adaboost.RT. (lines 176-177, 205-207)
- Define "weak learners" in the context of Adaboost.RT (Line 175).
Author Response: To avoid any confusion, we have replaced the term "weak learners" with "base learners" in the revised manuscript. (line 179)
- Define better why use Kernel Ridge Regression and Adaboost.RT. Make a brief introduction in the introduction section and justify why you are going to do this comparison. The rationale beyond the use these 3 methodologies is not clearly explained.
Author Response: Thank you for your suggestion. We have clarified the rationale for choosing machine learning methods in the introduction section. (lines 75-83)
- GWAS (Lines 216–226):
- Justify the choice of 50 kb as the optimal linkage distance (Line 225).
Author Response: Thank you for your comment. We have added a justification for the choice of 50 kb as the optimal linkage distance in the revised manuscript. (lines 232-234)
- As for Gibbs fixed term combination should be revised.
Author Response: We have reviewed the GWAS section but were unable to identify any mention of "Gibbs fixed term combination." Could you kindly provide further clarification or specify the part of the manuscript where this term is referenced?
Results
- Genetic Parameters (Lines 247–262):
- The high phenotypic correlation (>0.5) among traits is significant. Discuss its implications for selection strategies.
Author Response: Thank you for pointing this out. We have added the discussion regarding the high phenotypic correlation (>0.5) among traits and its implications for selection strategies in the discussion section of the manuscript. (lines 373-374)
- Genomic Prediction (Lines 267–285):
- The claim that ML methods improve GEBV accuracy by 22% and 11% (Line 278) is promising. Include confidence intervals to support these results.
Author Response: Thank you for your reminder. We have included the confidence intervals to support the results. (lines 289-297)
- GWAS (Lines 286–318):
- The identification of candidate genes is a major strength. However, the discussion on functional roles (e.g., Lines 305–317) lacks depth. Add more biological context or references.
Author Response: Thank you for your reminder. We have added relevant references in the discussion section and provided explanations for the candidate genes. (lines 403-423)
Discussion
- Heritability Estimates (Lines 334–344):
- Compare the heritability estimates with similar studies on goats or other livestock to contextualize findings and report in the text for better readability.
Author Response: Thank you very much for your comments. We have compared the heritability estimates with similar studies on goats and other livestock. (lines 355-357)
- The impact of environmental factors (Line 387) is mentioned briefly. Suggest conducting future studies to isolate these effects.
Author Response: We appreciate the suggestion. As recommended, we will consider conducting future studies to better isolate and analyze the effects of these factors on the traits of interest.
- Machine Learning (Lines 345–356):
- Discuss the computational cost of ML methods compared to traditional models like GBLUP. This is critical for practical applications.
Author Response: Thank you for your reminder. We have revised it in the updated manuscript. (lines 394-399)
- Candidate Genes (Lines 357–379):
- The functional roles of genes like GJB5 and PKN2 (Line 360) are intriguing but speculative. Suggest experimental validation for future studies.
Author Response: Thank you for your valuable suggestion. We will consider designing experiments to validate these genes in future studies.
Figures and Tables
- Figure 1 (Page 3):
- Add clear labels or annotations for each udder trait to improve comprehension.
Author Response: Thank you for your suggestion. We have implemented the revisions accordingly. (lines 110-113)
- Figure 3 (Page 8):
- The box plots are informative but lack clarity on error margins. Include confidence intervals or standard deviations.
Author Response: We greatly appreciate your suggestion. To maintain the simplicity of the figures, we have added the information on standard deviations and confidence intervals to the results section of the manuscript. (lines 289-297)
- Figure 4 (Page 9):
- The Manhattan plots are visually clear but lack a legend explaining the significance threshold.
Author Response: Actually, we selected the suggestive significance threshold, which is approximated to be around P = 10−6. This allows for a more balanced approach to identifying potential genetic associations.
- Table 2 (Page 7):
- The use of standard errors in parentheses is appropriate. However, explain this format in the legend for clarity.
Author Response: Thank you for your suggestion. We have explained this format in lines 273-277, which correspond to the legend of Table 2.
Typos and Errors
- Line 323: "suggested as" → "suggested to be."
Author Response: Thank you for your suggestion. I've updated 'suggested as' to 'suggested to be' on Line 339 as recommended.
- Line 381: "significant contributions to" → "significant contribution toward."
Author Response: Thank you for your reminder. I've updated ' significant contributions to ' to ' significant contribution toward' on Line 432 as recommended.
Reviewer 2 Report
Comments and Suggestions for Authors
Genomic and phenotypic evaluation methods enable accurate and efficient assessment of udder traits in dairy goats, empowering farmers to make more informed breeding decisions for animal selection. In this regard, the researchers have chosen the right topic to study the genomic prediction of udder traits in dairy goats. Upon reviewing the article, I found some suggestions to improve the study.
1. While studying udder traits, milk production traits are one of the major components, as udder traits are highly correlated with milk yield. The suggested article may consider including milk production traits as one of the parameters for their study.
2. From the results of the manuscript, the heritability estimates for udder traits ranged from low to moderate, spanning between 0.13 and 0.32. This does not show significant differences compared to conventional heritability estimations. Usually, GBLUP using GWAS data provides higher heritability values.
3, Furthermore, among the heritability values, the range between 0.13 (udderC) and 0.32 (udderB) shows considerable variation, which has not been explained properly. Additionally, the variance components have not been provided in the results nor adequately discussed.
Author Response
Comments and Suggestions for Authors
Genomic and phenotypic evaluation methods enable accurate and efficient assessment of udder traits in dairy goats, empowering farmers to make more informed breeding decisions for animal selection. In this regard, the researchers have chosen the right topic to study the genomic prediction of udder traits in dairy goats. Upon reviewing the article, I found some suggestions to improve the study.
- While studying udder traits, milk production traits are one of the major components, as udder traits are highly correlated with milk yield. The suggested article may consider including milk production traits as one of the parameters for their study.
Author Response: Thank you for your suggestion. Currently, we face some limitations in data collection, making it challenging to obtain individual milk yield data for each goat. However, we highly value this aspect and will make efforts to collect milk yield-related data in the future to enhance our study.
- From the results of the manuscript, the heritability estimates for udder traits ranged from low to moderate, spanning between 0.13 and 0.32. This does not show significant differences compared to conventional heritability estimations. Usually, GBLUP using GWAS data provides higher heritability values.
Author Response: We greatly appreciate your suggestion. We have carefully compared our findings with existing studies and observed that the heritability estimates for udder traits in dairy goats reported in our research are generally consistent with those documented in the literature. This alignment suggests that our results are within the expected range based on previous research. (lines 355-357)
3, Furthermore, among the heritability values, the range between 0.13 (udderC) and 0.32 (udderB) shows considerable variation, which has not been explained properly. Additionally, the variance components have not been provided in the results nor adequately discussed.
Author Response: We greatly appreciate your suggestion. We have now included the variance components in the results and provided a detailed discussion on them. (Lines 259-264, lines 357-364)
Reviewer 3 Report
Comments and Suggestions for Authors
Dear Author’s,
I thoroughly enjoyed reading your work, which provides valuable insights into genomic prediction and the genetic architecture of udder traits in Saanen dairy goats. While the manuscript is well-structured and presents valuable findings, several areas need further elaboration and refinement to enhance clarity, reproducibility, and practical relevance. Addressing the limitations and expanding on the implications of the findings will ensure the study's maximum impact in the field of livestock genetics.
Your study demonstrates the innovative use of low-coverage whole-genome sequencing (lcWGS) and advanced machine learning (ML) methods to address challenges in genomic prediction. Additionally, the identification of candidate genes associated with udder traits is a notable contribution to the field.
To strengthen your manuscript and enhance its impact, I recommend addressing the following points:
Provide more details about the computational requirements for GLIMPSE2, including processing time and hardware specifications.
Discuss the trade-offs associated with the sequencing depth of 1.95× and its influence on imputation accuracy and the reliability of downstream analyses.
Expand on how moderate heritability estimates (0.13–0.32) align with breeding goals for udder traits. Highlight how heritability impacts the expected genetic gains in selection programs.
Discuss potential environmental influences on trait variability and whether these were adequately controlled in the study.
Elaborate on why ML methods (KRR and Adaboost.RT) outperformed GBLUP, potentially linking these results to the complexity of genetic architecture or non-linear relationships among SNPs.
Discuss the practical implications of using ML methods in breeding programs, including computational challenges and scalability.
While the identification of genes such as GJB5, PKN2, and NAV3 is a significant finding, propose experimental approaches for validating their roles in udder trait development.
Discuss how the identified genes can be used to develop genomic selection indices or targeted breeding strategies.
Include details on the software and parameters used for analyses (e.g., imputation, genomic prediction) to ensure reproducibility.
Discuss how your findings can be translated into practical applications in dairy goat breeding programs.
Your work addresses an important area in livestock genetics and has significant implications for improving productivity and efficiency in the dairy goat industry. By addressing the points above, your manuscript will be even more impactful and informative for readers.
Best regards,
Author Response
Comments and Suggestions for Authors
To strengthen your manuscript and enhance its impact, I recommend addressing the following points:
Provide more details about the computational requirements for GLIMPSE2, including processing time and hardware specifications.
Author Response: Thank you for your suggestion. We have provided the details regarding the computational requirements for GLIMPSE2. (lines 133-135)
Discuss the trade-offs associated with the sequencing depth of 1.95× and its influence on imputation accuracy and the reliability of downstream analyses.
Author Response: Thank you for your suggestion. We have already discussed the trade-offs associated with the sequencing depth of 1.95× and its impact on imputation accuracy and the reliability of downstream analyses in the manuscript. (lines 340-343)
Expand on how moderate heritability estimates (0.13–0.32) align with breeding goals for udder traits. Highlight how heritability impacts the expected genetic gains in selection programs.
Author Response: Thank you for your valuable suggestion. We have already included a detailed discussion on how moderate heritability estimates (0.13–0.32) align with breeding goals for udder traits and the impact of heritability on expected genetic gains in selection programs. (lines 368-370)
Discuss potential environmental influences on trait variability and whether these were adequately controlled in the study.
Author Response: Thank you for your suggestion. We have already included a discussion on the potential environmental influences on trait variability and addressed whether these factors were adequately controlled in the study. (lines 360-364)
Elaborate on why ML methods (KRR and Adaboost.RT) outperformed GBLUP, potentially linking these results to the complexity of genetic architecture or non-linear relationships among SNPs.
Author Response: We greatly appreciate your suggestion. This point has been addressed in our discussion. (lines 392-398)
Discuss the practical implications of using ML methods in breeding programs, including computational challenges and scalability.
Author Response: Thank you for your valuable suggestion. We have already covered the practical implications of using ML methods in breeding programs. (lines 396-399)
While the identification of genes such as GJB5, PKN2, and NAV3 is a significant finding, propose experimental approaches for validating their roles in udder trait development.
Author Response: Thank you for your suggestion. While we have not conducted experimental validation in this study, we have added a discussion on the experimental approaches that should be considered for future research to validate the roles of GJB5, PKN2, and NAV3 in udder trait development. (lines 426-432)
Discuss how the identified genes can be used to develop genomic selection indices or targeted breeding strategies.
Author Response: Thank you for your suggestion. We have included a discussion on how the identified genes can be used to develop genomic selection indices and targeted breeding strategies in the manuscript. (lines 420-423)
Include details on the software and parameters used for analyses (e.g., imputation, genomic prediction) to ensure reproducibility.
Author Response: Thank you for your suggestion. We have included details on the software and parameters used for analyses. (lines 133-135, 176-177, 205-207)
Discuss how your findings can be translated into practical applications in dairy goat breeding programs.
Author Response: Thank you for your suggestion. We have discussed how our findings can be translated into practical applications in dairy goat breeding programs in the manuscript. (lines 420-423)
Reviewer 4 Report
Comments and Suggestions for Authors
Dear authors, respectfully, it seems to me that this is a good manuscript. However, many points affect the quality of the manuscript. The main factor that affects the manuscript is the generic writing style that decreases the relevance of the text and the number of traits evaluated. Why only three udder traits? Many other traits could have been evaluated. Or are you writing another manuscript with three other traits? This is not recommended. To my point of view, it is necessary to add results of more traits. Some suggestions are below.
Simple summary: The simple summary is fine if I read it idea by idea; however, a complete reading of this topic results in a confusing understanding. This is because the first idea describes that the goal is to find the genome of goat udders and the following ideas, mainly in lines 20-24, describe methods to predict heritability. Finally, the last two lines support something different from the previous ideas. I don't know if you understand my observation. My suggestion is to rewrite this text making a logical sequence for the readers. Sometimes, by adding a word as a connector, ideas are unified.
Abstract: The abstract is very generic. It is important to keep in mind that this part of the manuscript is the first reading and the readers' interest or not depends on this. Improve it.
Lines 29-31: Reading the objective, I understood what the study is about. The simple summary is also complete, but it lacks connectors between ideas. Perhaps you can use this objective as a structure for the simple summary.
Lines 31-33: Why only three udder traits? Many other traits could have been evaluated. Or are you writing another manuscript with three other traits? This is not recommended. To my point of view, it is necessary to add results of more traits.
Line 34: Only three traits were evaluated; why describe heritabilities as a range when you can write down the value of each trait?
Lines 34-35: “strong and favorable”: Avoid informal words, because strong may not be strong and favorable may not be favorable from the point of view of different readers. Add the values.
Lines 35-36: This text is repetitive with previous descriptions in the abstract. Remove it.
Lines 36-38: “strong” – looking at the values ​​found for heritability, I am not agree with “strong”.
Lines 38-41: Why only three models? There are many models available.
Line 44: 20 and 11, respectively? Complete or rewrite it.
Lines 45-47: Is this conclusion based on heritability values ​​20 and 11 improvement? This conclusion is very generic.
Lines 47-49: Repetitive. Remove it.
Keyword: My suggestion is to use keywords other than the title and review the keyword suggestion in the author's instructions.
Introduction: I like this introduction because it has relevant information. My suggestion is to add numbers to improve your description. Only as an example: Milk and other goat by-products generate up to 70% of income for producers.
Line 77: What does “ML” mean? Abbreviations should be described the first time they appear. Also, avoid starting a paragraph with abbreviations. Consider the abstract and the body of the manuscript as different manuscripts where abbreviations should be described in detail the first time they are used.
Line 83: Line 77: What does “GEBVs” mean?
Line 85: Line 77: What does “GBLUP” mean?
Material and methods
Lines 107-108: Describe this part specifically, in more detail.
Line 109: Similar to the previous comment.
Lines 109-110: Similar to the previous comment.
Results. The results description is generic. Improve your writing style showing your data in other forms more than a simple description of tables. Was higher? How much (%, g, l, etc)?
Lines 252-253: Is this statement correct? I don't understand. You evaluate 3 traits and the correlation between them and the conclusion is that these 3 traits essentially describe the udder? How, if you don't evaluate many other traits?
Table 2: I am sure that for authors, the descriptions “UdderA, udderB and udderC” are acceptable; however, for readers, it is tiring to remind them again what each one is. Please use a real description like “udder height” throughout the text instead of udderA….
Discussion: The topic of discussion is very speculative. The discussion should focus on explaining how the results were obtained. For this, add theories, hypotheses or statements about how you obtained your results, whether biologically, metabolically, physiologically, environmentally, etc. In the current situation, the discussion is a good general review and comparison of data with other authors; however, you need to make a specific description of how the results were obtained.
Line 352: “robust reliability” – Add a value to this statement and do the same for all generic descriptions that could be improved by adding numbers.
Line 355: “excellent reliability” – Add a value to this statement and do the same for all generic descriptions that could be improved by adding numbers.
Line 375: “breast cancer” in goats? Thinking about this point, is the above description of the genes a description of the functionality of these genes in goats? Or in other species? One must be careful with that because we know that genes can have different effects in different species.
Conclusion: Repeat the objective of the study on lines 396-398 is not necessary. Be direct to the conclusion.
I am not agree with the conclusion on lines 399-400 because don’t models evaluation were added. Perhaps my observation is wrong but that is why the conclusion needs to be more objective and descriptive.
Author Response
Comments and Suggestions for Authors
Dear authors, respectfully, it seems to me that this is a good manuscript. However, many points affect the quality of the manuscript. The main factor that affects the manuscript is the generic writing style that decreases the relevance of the text and the number of traits evaluated. Why only three udder traits? Many other traits could have been evaluated. Or are you writing another manuscript with three other traits? This is not recommended. To my point of view, it is necessary to add results of more traits. Some suggestions are below.
Author Response: Thank you for your suggestion. Due to current resource constraints, we were unable to include other measurements in the present study. However, we plan to incorporate other udder traits, like some nipples measurement and angles, in future research as our resources allow.
Simple summary: The simple summary is fine if I read it idea by idea; however, a complete reading of this topic results in a confusing understanding. This is because the first idea describes that the goal is to find the genome of goat udders and the following ideas, mainly in lines 20-24, describe methods to predict heritability. Finally, the last two lines support something different from the previous ideas. I don't know if you understand my observation. My suggestion is to rewrite this text making a logical sequence for the readers. Sometimes, by adding a word as a connector, ideas are unified.
Author Response: We thank the reviewer for pointing this out. We have rewritten this paragraph to make it more logical. (lines 17-27)
Abstract: The abstract is very generic. It is important to keep in mind that this part of the manuscript is the first reading and the readers' interest or not depends on this. Improve it.
Author Response: Thank you for your reminder. We have rewritten and improved the abstract, please check it. (lines 28-42)
Lines 29-31: Reading the objective, I understood what the study is about. The simple summary is also complete, but it lacks connectors between ideas. Perhaps you can use this objective as a structure for the simple summary.
Author Response: Thank you for your suggestion. We have rewritten and improved the abstract, please check it. (lines 28-42)
Lines 31-33: Why only three udder traits? Many other traits could have been evaluated. Or are you writing another manuscript with three other traits? This is not recommended. To my point of view, it is necessary to add results of more traits.
Author Response: Due to current resource constraints, we were unable to include other measurements in the present study. However, we plan to incorporate other udder traits, like some nipples measurement and angles, in future research as our resources allow.
Line 34: Only three traits were evaluated; why describe heritabilities as a range when you can write down the value of each trait?
Author Response: Thank you for your suggestion. We have provided the heritability estimates for each trait. (line 34)
Lines 34-35: “strong and favorable”: Avoid informal words, because strong may not be strong and favorable may not be favorable from the point of view of different readers. Add the values.
Author Response: Thank you for your careful revision. We have added the values in line 35.
Lines 35-36: This text is repetitive with previous descriptions in the abstract. Remove it.
Author Response: Thank you for your careful revision. We have removed this text and rewritten the abstract. (lines 28-42)
Lines 36-38: “strong” – looking at the values ​​found for heritability, I am not agree with “strong”.
Author Response: Thank you for your suggestion. We have rewritten this sentence in the revised manuscript. (lines 35-36)
Lines 38-41: Why only three models? There are many models available.
Author Response: The selection of three models was based on the data available to us, which lacks pedigree information. Given this limitation, GBLUP was considered the most appropriate model for our analysis. Furthermore, we included machine learning models to assess their potential advantages in predicting the traits of interest and to evaluate whether they might offer more accurate or reliable predictions compared to GBLUP in the context of our study.
Line 44: 20 and 11, respectively? Complete or rewrite it.
Author Response: Thanks for your kindly reminder. We have rewritten this sentence in line 39.
Lines 45-47: Is this conclusion based on heritability values ​​20 and 11 improvement? This conclusion is very generic.
Author Response: Thank you for your suggestion. We have rewritten this sentence in the revised manuscript, please check it. (lines 38-40)
Lines 47-49: Repetitive. Remove it.
Author Response: Thank you for your careful work. We have removed it.
Keyword: My suggestion is to use keywords other than the title and review the keyword suggestion in the author's instructions.
Author Response: Thank you for your suggestion. We have revised the keyword in the updated manuscript. (lines 43-44)
Introduction: I like this introduction because it has relevant information. My suggestion is to add numbers to improve your description. Only as an example: Milk and other goat by-products generate up to 70% of income for producers.
Author Response: Thank you very much for your comments. We have added numbers to improve the description. (lines 48-49)
Line 77: What does “ML” mean? Abbreviations should be described the first time they appear. Also, avoid starting a paragraph with abbreviations. Consider the abstract and the body of the manuscript as different manuscripts where abbreviations should be described in detail the first time they are used.
Author Response: Thank you for your suggestion. We have addressed this issue and made the necessary corrections. (line 69)
Line 83: Line 77: What does “GEBVs” mean?
Author Response: Thank you for your suggestion. We have provided an explanation for "GEBVs" in the revised manuscript. (line 75)
Line 85: Line 77: What does “GBLUP” mean?
Author Response: Thank you for your suggestion. We have provided an explanation for "GBLUP" in the revised manuscript. (line 78)
Material and methods
Lines 107-108: Describe this part specifically, in more detail.
Author Response: Thank you very much for your comments. We have provided a more detailed description of this part in the revised manuscript. (lines 105-108)
Line 109: Similar to the previous comment.
Author Response: Thank you for your suggestion. We have revised it in line 108.
Lines 109-110: Similar to the previous comment.
Author Response: Thank you for pointing this out. We have revised it in lines 107-108.
Results. The results description is generic. Improve your writing style showing your data in other forms more than a simple description of tables. Was higher? How much (%, g, l, etc)?
Author Response: Thank you for your suggestion. We have improved the writing style showing our data more specific. (lines 259-264; 289-297)
Lines 252-253: Is this statement correct? I don't understand. You evaluate 3 traits and the correlation between them and the conclusion is that these 3 traits essentially describe the udder? How, if you don't evaluate many other traits?
Author Response: Thank you for your suggestion. We have revised this sentence in lines 265-266.
Table 2: I am sure that for authors, the descriptions “UdderA, udderB and udderC” are acceptable; however, for readers, it is tiring to remind them again what each one is. Please use a real description like “udder height” throughout the text instead of udderA….
Author Response: I appreciate your recommendation. We have made the necessary revisions throughout the text and replaced "UdderA," "udderB," and "udderC" with more specific descriptions.
Discussion: The topic of discussion is very speculative. The discussion should focus on explaining how the results were obtained. For this, add theories, hypotheses or statements about how you obtained your results, whether biologically, metabolically, physiologically, environmentally, etc. In the current situation, the discussion is a good general review and comparison of data with other authors; however, you need to make a specific description of how the results were obtained.
Author Response: Thank you for your suggestion. We have revised the discussion section to provide a more specific explanation
Line 352: “robust reliability” – Add a value to this statement and do the same for all generic descriptions that could be improved by adding numbers.
Author Response: We thank the reviewer for pointing this out. We have added the value to this statement in lines 387-389, and also improved this manuscript. (lines 355-364; 373-374; 426-430)
Line 355: “excellent reliability” – Add a value to this statement and do the same for all generic descriptions that could be improved by adding numbers.
Author Response: Thank you for your suggestion. We have added the value to this statement in lines 389-391.
Line 375: “breast cancer” in goats? Thinking about this point, is the above description of the genes a description of the functionality of these genes in goats? Or in other species? One must be careful with that because we know that genes can have different effects in different species.
Author Response: Thank you for your suggestion. We have revised this statement in the updated manuscript. (lines 418-420)
Conclusion: Repeat the objective of the study on lines 396-398 is not necessary. Be direct to the conclusion.
Author Response: Thank you for your reminder. We have removed this sentence in the revised manuscript. (line 448)
I am not agree with the conclusion on lines 399-400 because don’t models evaluation were added. Perhaps my observation is wrong but that is why the conclusion needs to be more objective and descriptive.
Author Response: We thank the reviewer for pointing this out. We have made revisions to this sentence in lines 450-453.
Round 2
Reviewer 1 Report
Comments and Suggestions for Authors
Dear Author, Thank you for making the suggested corrections to the manuscript. As far as I can judge it is fit for publication in its present form.
Reviewer 2 Report
Comments and Suggestions for Authors
The authors' comments are accepted.
Reviewer 3 Report
Comments and Suggestions for Authors
The authors have improved the manuscript and it is now ready for publication. I have no further comments.
Sincerely,
Reviewer 4 Report
Comments and Suggestions for Authors
Dear authors, reviewers make suggestions with the aim of improving the manuscript; however, it is the responsibility of the authors to accept them in full, accept them partially or reject them. I am satisfied with the responses.